# *Aspergillus fumigatus* Supernatants Disrupt Bronchial Epithelial Monolayers: Potential Role for Enhanced Invasion in Cystic Fibrosis

**DOI:** 10.3390/jof9040490

**Published:** 2023-04-19

**Authors:** Katie Dunne, Emma Reece, Siobhán McClean, Sean Doyle, Thomas R. Rogers, Philip Murphy, Julie Renwick

**Affiliations:** 1Discipline of Clinical Microbiology, School of Medicine, Trinity College Dublin, D02 PN40 Dublin, Ireland; 2School of Biomolecular and Biomedical Science, University College Dublin, Belfield, D04 V1W8 Dublin, Ireland; 3Department of Biology, Maynooth University, Maynooth, W23 F2K8 Kildare, Ireland

**Keywords:** *Aspergillus fumigatus*, inflammation, gliotoxin, supernatants, transepithelial electrical resistance, tight junctions, zonula occludens-1, junctional adhesion molecule-A, cystic fibrosis

## Abstract

*Aspergillus fumigatus* is the most commonly isolated fungus in chronic lung diseases, with a prevalence of up to 60% in cystic fibrosis patients. Despite this, the impact of *A. fumigatus* colonisation on lung epithelia has not been thoroughly explored. We investigated the influence of *A. fumigatus* supernatants and the secondary metabolite, gliotoxin, on human bronchial epithelial cells (HBE) and CF bronchial epithelial (CFBE) cells. CFBE (F508del CFBE41o^−^) and HBE (16HBE14o^−^) trans-epithelial electrical resistance (TEER) was measured following exposure to *A. fumigatus* reference and clinical isolates, a gliotoxin-deficient mutant (Δ*gliG*) and pure gliotoxin. The impact on tight junction (TJ) proteins, zonula occludens-1 (ZO-1) and junctional adhesion molecule-A (JAM-A) were determined by western blot analysis and confocal microscopy. *A. fumigatus* conidia and supernatants caused significant disruption to CFBE and HBE TJs within 24 h. Supernatants from later cultures (72 h) caused the greatest disruption while Δ*gliG* mutant supernatants caused no disruption to TJ integrity. The ZO-1 and JAM-A distribution in epithelial monolayers were altered by *A. fumigatus* supernatants but not by Δ*gliG* supernatants, suggesting that gliotoxin is involved in this process. The fact that Δ*gliG* conidia were still capable of disrupting epithelial monolayers indicates that direct cell–cell contact also plays a role, independently of gliotoxin production. Gliotoxin is capable of disrupting TJ integrity which has the potential to contribute to airway damage, and enhance microbial invasion and sensitisation in CF.

## 1. Introduction

*Aspergillus fumigatus* is the most commonly isolated fungal pathogen in chronic lung diseases and particularly in those individuals with cystic fibrosis (CF) [1,2,3,4]. CF is a life-threatening hereditary disorder that affects multiple organs of the body, with the respiratory system most severely affected. Mutations in the CF transmembrane conductance regulator (CFTR) gene result in the absence of functioning CFTR proteins causing decreased chloride secretions and increased sodium absorption at the apical membrane of the airway epithelial cells leading to abnormally viscous mucous [5]. Coupled with impaired mucociliary clearance, this makes the CF airways an ideal environment for persistent microbial colonisation. Airway disease in CF is characterised by a continuous cycle of infection and inflammation, playing a significant role in exacerbations and disease progression [6,7]. The most common manifestation of *A. fumigatus* in CF is allergic bronchopulmonary aspergillosis (ABPA) occurring in 1–15% of CF patients and causing reduced pulmonary function [8]. Up to 60% of CF patients are colonised with *Aspergillus* [9,10,11,12] and we have shown that CF patients airways can be persistently colonised with an indistinguishable genotype of *A. fumigatus* for several months (Appendix A). Persistent *A. fumigatus* has recently been associated with adverse clinical outcomes, such as lung function decline [13,14,15,16] and more frequent exacerbations [13,14,15]. Patients with non-ABPA *Aspergillus* colonisation have shown lung function stabilisation and reduced exacerbations following treatment with itraconazole, suggesting a pathogenic role for *A. fumigatus* in CF [4].

The CF airways are host to a diverse microbial community made up of common CF bacterial species such as *Pseudomonas aeruginosa*, *Staphylococcus aureus*, *aureus*, *Haemophilus influenzae*, *Stenotrophomonas maltophilia*, non-tuberculous *Mycobacteria* and *Burkholderia* species [17,18], and many anaerobic species now identified by culture-independent methods [18,19]. *A. fumigatus* commonly co-colonises the airways with *P. aeruginosa*. Co-colonisation with these two microorganisms results in reduced lung function, an increased number of hospitalisations and respiratory exacerbations and increased usage of antimicrobials relative to either pathogen colonising alone [20]. In light of the recent pandemic, the clinical significance of co-infections has been further highlighted. COVID-19-associated pulmonary aspergillosis (CAPA) has been shown to impact 13.5% (range 2.5–35.0%) of COVID-19 patients [21]. The severity of the complications of COVID-19 were shown to increase with a coinfection of fungi [22] and the mortality rates of COVID-19 patients with aspergillosis have been reported to be as high as 44% [23].

The ability of *A. fumigatus* to compromise the airway barrier could have significant implications in mixed microbial infections. The epithelial cells of the airways are tightly bound together providing a highly regulated and impermeable barrier formed by tight junctions (TJs) [24,25]. The TJs regulate ions, water and immune cell transport between the epithelial cells of the airways and prevent microbial translocation [24]. The TJs are formed by a complex of different proteins including zonula occludens 1 (ZO-1), junctional adhesion molecule A (JAM-A), occludins and claudins, each having roles in the formation and regulation of the TJs [25]. CF bacterial pathogens such as *Burkholderia cenocepacia* have been shown to penetrate the epithelial barrier in vitro by altering TJ proteins [26]. Compromised epithelial barriers may result in the mass infiltration of neutrophils and an invasion of microorganisms, which can have severe consequences in people with chronic lung diseases.

Studies have been published investigating the effects of *A. fumigatus* supernatants on airway epithelia [27]; however, few investigate the effects of early supernatants and there are no studies investigating the effects of *A. fumigatus* supernatants on CF epithelia. *A. fumigatus* has a diverse secretome, producing several proteases, allergens and toxins [28]. One such mycotoxin, gliotoxin, has been shown to have numerous immunoevasive effects [29,30]; however, whether gliotoxin affects the integrity of lung epithelial barriers remains unclear. Here, we investigated the effects of *A. fumigatus* isolates and their supernatants on the integrity of bronchial epithelial cell monolayers with (16HBE14o^−^) and without functioning CFTR (F508del CFBE41o^−^) and demonstrate that gliotoxin disrupts the structure of the epithelial monolayer.

## 2. Materials and Methods

### 2.1. Microbiological Strain Maintenance, Storage and Culture Conditions

*A. fumigatus* strain Af293 (ATCC^®^ MYA-4609^TM^), a gliotoxin deficient mutant Δ*gliG* derived from the Af293 wild type [31] and two clinical *A. fumigatus* strains isolated from CF sputum samples from two different patients, namely AF1 and AF2, were all maintained on Malt Extract Agar (MEA) (Fannin, Dublin, Ireland). Stocks of all *A. fumigatus* isolates were kept in 15% glycerol (*v*/*v*) at −80 °C. The Δ*gliG* mutant was generated using the bipartite marker technique and is deficient in producing gliotoxin [31,32]. The AF1 strain was isolated from 6 consecutive sputum samples from a single CF patient taken over the course of 6 months and confirmed using the STRAf assay [33] to be an indistinguishable genotype (Appendix A). The AF2 strain was isolated from a single sputum sample taken from a CF patient and not subsequently isolated from the same patient (Appendix A).

### 2.2. Maintenance of CFBE and Human Bronchial Epithelial (HBE) Cell Lines

A lung epithelial cell line, CFBE41o^−^ (CFBE), collected from a female CF patient homozygous for the CFTR delF_508_ mutation and a non-CF human bronchial epithelial cell line 16HBE14o^−^ [34] were used in this study (gift from Dr Dieter Greunert, UCSF). The CFBE cells (passage P3.103 to P3.123) and HBE cells (passage P2.77 to P2.97) were maintained at 37 °C with 5% CO_2_, in fibronectin coated flasks in minimal essential medium supplemented (S-MEM) with 1% L-glutamine (*v*/*v*), 1% penicillin/streptomycin (*v*/*v*), 10% foetal bovine serum (*v*/*v*) and 1% non-essential amino acids [26]. Cells were subcultured by trypsin digestion (when 60–80% confluence was reached) into sterile fibronectin coated T-75 flasks (Merck, Rahway, NJ, USA) and incubated at 37 °C with 5% CO_2_.

### 2.3. Infection of CFBE and HBE Monolayers with A. fumigatus Isolates

For tight junction assays, CFBE or HBE cell lines were seeded onto each 0.4 μm polyester transwell filter (Merck) at a dilution of 1 × 10^5^/0.5 mL with 1.5 mL of media applied to the basolateral side of the transwell support. A blank well containing only S-MEM [26] and a control with cells only were also included. Following 18 to 24 h incubation, the apical medium was removed creating an air–liquid interface [26]. S-MEM was replaced on alternate days and cells were cultured over a 5-day period to enable the formation of TJs. On day 6, Trans-epithelial electrical eesistance (TEER) measurements were taken in triplicate for all wells using an EVOM meter (World Precision Instruments) and only those exceeding TEER measurement of 150 Ω·cm^2^ (above blank) were used for the experiment. CFBE and HBE monolayers were infected with Af293, Δ*gliG*, AF1 and AF2 conidia at a multiplicity of infection (MOI) of 2:1 and incubated at 37 °C with 5% CO_2_. Sterile latex beads (Merck) 3 µm in diameter (MOI 2:1) were also seeded onto CFBE and HBE monolayers as controls. TEER values were recorded at 0, 2, 4, 6, 8, 10, 12 and 24 h post infection. Experiments were performed on three independent occasions. Triplicate ten-fold dilutions of conidia concentrations were made for plating to confirm conidia concentration of MOI for each experiment.

TEER values were calculated using the following formula:
$$\frac{\mathrm{Resistance}\;\mathrm{of}\;\mathrm{cells}\;\mathrm{on}\;\mathrm{filter}–\mathrm{resistance}\;\mathrm{of}\;\mathrm{blank}\;\mathrm{filter}\;(\mathrm{with}\;\mathrm{no}\;\mathrm{cells})}{1.12\;{\mathrm{cm}}^{2}\;(\mathrm{area}\;\mathrm{of}\;\mathrm{filter})}=\mathrm{TEER}\;\Omega \cdot {\mathrm{cm}}^{2}$$


Results were recorded as a percentage of the control TEER for each time point.

### 2.4. Treatment of CFBE and HBE Monolayers with Supernatants from A. fumigatus Isolates

Seven cultures containing 2 × 10^5^ conidia/3 mL minimal essential medium (MEM) (Merck) for each of the four isolates were incubated at 37 °C for 0, 4, 8, 12, 24, 48 or 72 h. Cultures were passed through a sterile cell sieve and then sterile filtered using a 0.2 µm syringe filter (Sarstedt, Lower Saxony, Germany) to remove all *A. fumigatus* conidia and hyphae. Supernatants were confirmed as containing no viable conidia or hyphae by plating 20 µL of the filtered supernatants onto an MEA plate in triplicate and incubated at 37 °C for 7 days. No growth confirmed sterile filtration of supernatants. CFBE and HBE monolayers with TEER measurements exceeding 150 Ω·cm^2^ were inoculated with 0, 4, 8, 12, 24, 48 or 72 h supernatants of the four *A. fumigatus* strains. TEERs were monitored every 2 h from 0 to 12 h and again at 24 h post infection. Experiments were performed on three independent occasions.

### 2.5. Exposure of CFBE and HBE Monolayers to Gliotoxin

Gliotoxin (Merck) was suspended in DMSO at a stock concentration of 0.1 M and was diluted to working solutions in MEM. All final gliotoxin concentrations contained no more than 0.08% DMSO for tissue culture assays. Gliotoxin was applied to CFBE and HBE monolayers at a concentration of 0.8, 8 or 80 µM. Controls of untreated monolayers and 0.08% DMSO-treated monolayers were included. TEERs were monitored every two hours from 0 to 12 h and again at 24 h post infection. Experiments were performed on three independent occasions.

### 2.6. Western Blot Analysis of TJ Proteins JAM-A and ZO-1

CFBE and HBE monolayers were prepared as previously described. Supernatants of Af293 and Δ*gliG* grown for 4, 8 or 72 h were prepared as previously described and were applied to transwell inserts containing monolayers. For all experiments, a cell-only control was included. Following 24 h exposure, epithelial cell monolayers were washed three times with ice cold PBS and protein was harvested by adding 100 µL radioimmunoprecipitation assay buffer (RIPA) (Merck) plus protease inhibitors (Roche, Basel, Switzerland) to each transwell filter. Cells were gently scraped off using a sterile cell scraper and lysed with repeated pipetting. The suspension was transferred to a fresh tube on ice and incubated for at least 15 min. Samples were then sonicated for 1 min and transferred immediately to −80 °C. Proteins were quantified using the Qubit fluorometer and protein quantification assay kit (Life Technologies, Agarwal City Mall, Delhi, India). Protein samples (30 µg/ 20 µL) were loaded on 4–12% Bis-Tris SDS gels (NU-PAGE, Life Technologies) and separated by electrophoresis alongside a 4–250 kDa protein standard (Life Technologies) at 200 V for 50 min. Proteins were then transferred to nitrocellulose membranes at 30 V for 1 h. Membranes were blocked for 1 h with Tris-buffered saline with 5% (*w*/*v*) non-fat dried milk and 0.1% (*v*/*v*) Tween 20 (M-TBST) for glyceraldehyde 3-phosphate dehydrogenase protein (GAPDH) or with 5% (*w*/*v*) bovine serum albumin (BSA) and 0.1% (*v*/*v*) Tween 20 (BSA-TBST) for ZO-1 and JAM-A. Blots were then incubated with primary antibody (1:5000 for GAPDH (Calbiochem, San Diego, CA, USA) and (1:500 for ZO-1 (Life Technologies) and JAM-A (Santa Cruz Biotechnology, Dallas, TX, USA) over night at 4 °C. Membranes were washed and then incubated with either horseradish peroxidase (HRP)-conjugated anti-mouse (1:50,000 for GAPDH and JAM-A) or HRP-conjugated anti-rabbit (1:50,000 for ZO-1) (Thermoscientific, Waltham, MA, USA) for 1 h at room temperature. Following immunoblotting, proteins were detected by chemiluminescence and visualised using a Fuji film ImageQuant LAS 4000 analyser. The density of each band was compared with the corresponding control band and normalised against GAPDH by densitometry using Quantity One version 4.6.6 software. Results are presented as percentage of the untreated control.

### 2.7. Immuno-Fluorescence and Confocal Microscopy of CFBE and HBE Monolayers Exposed to Supernatants of Af293 or ΔgliG

CFBE and HBE monolayers were grown on transwell inserts as previously described. After 24 h exposure to medium alone or supernatants of Af293 or Δ*gliG*, monolayers were prepared for immuno-fluorescent imaging. Monolayers were washed with PBS, then permeablised with ice cold methanol (Merck) for 30 min and blocked with PBS containing 1% BSA (*w*/*v*) for 10 min. Cells were then incubated with 10 µg/mL primary antibody (rabbit anti-ZO-1 (Life Technologies) or mouse anti-JAM-A (Santa Cruz Biotechnology)) for 1 h at room temperature. Cells were then washed four times with PBS containing 1% BSA (*w*/*v*) and subsequently incubated with FITC-conjugated anti-rabbit (ZO-1) or anti-mouse (JAM-A) (Jackson Immunoresearch, West Grove, PA, USA) antibodies for 1 h at room temperature protected from the light. Monolayers were washed five times with PBS containing 1% BSA (*w*/*v*) and then post-fixed in PBS containing 4% paraformaldehyde (*w*/*v*) for 10 min. Following fixation, filters were removed from the inserts and mounted on slides with Vectashield^®^ containing DAPI (Vector Laboratories) and examined by confocal microscopy. Confocal images were captured at ×600 magnification using the Olympus FLOUVIEW FV1000 microscope utilising the FITC and DAPI channels and FV10-ASW software version 03.01. Images are representative of 3 fields of view taken per treatment.

### 2.8. Statistical Analysis

All statistical analysis was performed on GraphPad Prism version 5 and 9. Statistical analysis using the two-way ANOVA and Bonferroni post-test was carried out on each experiment investigating the ability of conidia or supernatants or gliotoxin to affect the TEER of CFBE and HBE monolayers at 2 h intervals over a 24 h period. Comparison of densitometry results from western blots was performed using one-way ANOVA and Bonferroni post-test. *p* values of <0.05 were considered significant.

## 3. Results

### 3.1. A. fumigatus Isolates and Their Supernatants Opened CFBE and HBE TJs

*A. fumigatus* conidia of the Af293 reference strain and CF clinical isolates (AF1 and AF2) disrupted CFBE TJs within 12 h (Figure 1A) and HBE TJs within 24 h (Figure 1B). A statistically significant decrease in the TEER values was observed between infected versus non-infected (control) CFBE monolayers from 8 h post-infection for Af293 (*p* < 0.01; two-way ANOVA) and from 12 h post-infection for both clinical isolates (*p* < 0.01; two-way ANOVA) (Figure 1A). In contrast, a significant decrease in the TEER values of HBE monolayers was only reached following 24 h infection (*p* < 0.01; two-way ANOVA) (Figure 1B). There was no significant difference in the ability of AF1 and AF2 conidia to impair CFBE or HBE TJs (Figure 1A). Untreated CFBE and HBE monolayers maintained TJ integrity, with TEER values ranging from 90% to 114% and 83% to 105% of the initial TEER for the duration of the experiments, respectively.

Supernatants of Af293, AF1 and AF2 were capable of opening the TJs of CFBE cells (Figure 1C). The 72 h supernatants of Af293 caused the greatest degree of disruption to TJ integrity. The 12 h and 24 h supernatants of the two clinical isolates were more disruptive to CFBE integrity than the corresponding supernatants of Af293 (*p* < 0.01; two-way ANOVA). In contrast, the later supernatants (48 h and 72 h) of Af293 were more disruptive to CFBE TJs than the later supernatants of the clinical isolates (*p* < 0.05; two-way ANOVA). The 72 h supernatant of AF2 was significantly more disruptive to TJ integrity than the 72 h supernatant of AF1 (*p* < 0.001). All isolate supernatants from the 4 h and 8 h cultures were capable of significantly disrupting TJ integrity within 24 h (*p* < 0.0001; two-way ANOVA) (Figure 2A). The disruption to TJ integrity by the 72 h supernatants of *A. fumigatus* was comparable in CFBE and HBE monolayers with a significant decrease in TEER occurring at 2 h post-exposure for both cell types (*p* < 0.0001; two-way ANOVA) (Figure 1D).

Overall, the 72 h supernatant of Af293 caused the greatest disruption to CFBE and HBE TJ integrity and this effect was induced as early as 2 h post-exposure to the supernatant (Figure 1D).

### 3.2. Supernatants from the Gliotoxin Deficient A. fumigatus Mutant, ΔgliG, Do Not Maintain the Ability of the Af293 Wild Type to Disrupt CFBE and HBE Monolayers

Conidia of the wild type Af293 and the gliotoxin mutant, Δ*gliG,* caused a significant decrease in the CFBE monolayer TEER compared with the control 24 h post-infection (*p* < 0.0001; two-way ANOVA) (Figure 2A). A similar trend was observed in HBE cells with a significant decrease in TEER occurring following 24 h exposure to conidia (*p* < 0.0001) (Figure 2B). As in CFBE cells, the conidia of Af293 caused significant disruption to the HBE monolayers from 10 h post-infection (*p* < 0.01) (Figure 2A,B) whereas conidia of Δ*gliG* took 24 h to cause significant disruption to monolayers.

Supernatants from Af293 culture time points as early as 4 h and 8 h were shown to disrupt CFBE TJs after 24h (Figure 1C); therefore, the effect of 4 h, 8 h and 72 h supernatants of Af293 and Δ*gliG* were compared. The 72 h supernatant of Af293 caused the greatest decrease in the CFBE TEER at 24 h (*p* < 0.0001; two-way ANOVA), while the 72 h supernatant of Δ*gliG* caused no significant decrease in TEER (Figure 2C). The 4 h and 8 h supernatants of Af293 and Δ*gliG* both caused comparable decreases in TEER at 24 h (*p* < 0.05; two-way ANOVA). These results were mirrored in HBE cells (Appendix A). To further investigate whether gliotoxin is capable of opening CFBE and HBE TJs, monolayers were treated with 0.8 µM, 8 µM and 80 µM pure gliotoxin (Sigma). All concentrations of gliotoxin caused significant reductions in CFBE and HBE monolayer TEER values at 24 h (*p* < 0.0001; one-way ANOVA) (Figure 2D). The 0.08% DMSO control (gliotoxin diluent) caused no significant reduction in TEERs.

### 3.3. TJ Proteins, ZO-1 and JAM-A, Are Disrupted by 72 h Supernatant of Af293 but Not the ΔgliG

TJs are specialised multi-protein complexes that define epithelial organisation. Two proteins with key roles in maintaining barrier function are ZO-1 and JAM-A. Thus, to further investigate the abilities of *A. fumigatus* supernatants to impair CFBE and HBE epithelial integrity, western blot analysis of two TJ proteins, ZO-1 and JAM-A, was performed. Although 4 h and 8 h supernatants of Af293 and Δ*gliG* were capable of opening TJs, no reduction in either ZO-1 or JAM-A in CFBE (Figure 3A) or HBE cells (Figure 3B) was observed in response to these supernatants. In contrast CFBE cells (Figure 3A) and HBE cells (Figure 3B) exposed to 72 h supernatants of Af293 showed reduced levels of ZO-1 and JAM-A TJ proteins. This was not observed in CFBE or HBE cells exposed to the 72 h supernatant of the gliotoxin deficient mutant, Δ*gliG.* The levels of ZO-1 and JAM-A detected in CFBE cells exposed to the 72 h supernatant of Af293 were significantly lower than in CFBE cells exposed to the 72 h supernatant of Δ*gliG* (*p* < 0.01; two-way ANOVA) (Figure 3A). This result was mirrored in HBE cells (Figure 3B). GAPDH levels in CFBE and HBE cells exposed to the 72 h supernatant of Af293 were also slightly reduced; therefore, the concentrations of JAM-A and ZO-1 levels were normalised to GAPDH in all the sample treatment scenarios and normalised change ZO-1 and JAM-A highlights the specificity of the effect on TJ proteins.

### 3.4. The 72 h Supernatant of Af293, but Not ΔgliG, Disrupts CFBE and HBE Monolayer TJ Physiology

FITC staining of ZO-1 and JAM-A TJ proteins in untreated CFBE cells (control) showed that ZO-1 (Figure 4A) and JAM-A (Figure 4B) are localised primarily in the plasma membranes of CFBE cells. As expected, JAM-A is localised mainly in the plasma membranes of cells but also localises in the cytoplasm of the CFBE cells. The staining of these proteins highlights the network of CFBE cells forming a monolayer and the honeycomb arrangement of TJ proteins, the hallmark of an intact epithelial monolayer.

CFBE monolayers exposed to 72 h supernatants of Af293 show a complete delocalisation of ZO-1 (Figure 4A) and JAM-A (Figure 4B). Instead, ZO-1 and JAM-A staining is present in the cytoplasm of some cells (arrow) and/ or in vesicles within the cell as demonstrated by punctate staining (arrow) with greatly reduced plasma membrane staining. In contrast, CFBE monolayers exposed to 72 h supernatants of the gliotoxin deficient mutant, Δ*gliG,* appeared similar to the untreated CFBE monolayers with evident networks of TJs. These results were again mirrored in the HBE monolayers (Figure 4C,D). These results strongly suggest the role of gliotoxin in TJ disruption and the destruction of the epithelial barrier.

## 4. Discussion

All *A. fumigatus* isolates were capable of disrupting CFBE TJ integrity by 12 h post-infection, with the reference strain opening the TJs as early as 8 h post-infection. Both clinical *A. fumigatus* isolates opened the CFBE TJs at a similar rate and all *A. fumigatus* isolates caused the same level of disruption to TJ integrity by 24 h. Amitani et al. identified conidia within the TJs of cells 6 h post-infection and hyphae penetrating epithelial cells and through the TJs 12 h post-infection of human bronchial mucosal tissue [35]. We observed initial disruptions of TJs of CFBE cells between 8 h and 12 h post-infection, corroborating previous findings in normal bronchial epithelial cells and giving a quantifiable measure of the disruption of CFBE TJs by *A. fumigatus* using TEER. Interestingly, conidia of the gliotoxin deficient mutant, Δ*gliG,* were equally capable of disrupting TJ integrity; however, this disruption was delayed. Both the wild type and Δ*gliG* mutant have been shown to cause similar levels of mortality in an insect infection model [31]. Davis et al. noted that there was no significant difference in the growth rates of the Δ*gliG* mutant and the wild type [31], ruling this out as a possible cause of the delayed TJ disruption by Δ*gliG* observed here.

Supernatants of all *A. fumigatus* isolates in this study were capable of disrupting CFBE TJ integrity. *A. fumigatus* supernatants as early as 4 h and 8 h were capable of significantly disrupting TJ integrity. The effects of these early supernatants are unlikely to be due to gliotoxin as the production of gliotoxin by *A. fumigatus* is known to be minimal before 24 h growth [36]. Additionally, we observed that even the 4 h and 8 h supernatants of the gliotoxin mutant were capable of disrupting TJs to a degree. Other groups have shown spore diffusates of *A. fumigatus* to inhibit phagocytosis by alveolar macrophages [37] and neutrophils [38]. These findings raise questions regarding the damage that early and short-term *A. fumigatus* colonisation may exert on the CF airways. The component/s of early *A. fumigatus* supernatants capable of disrupting TJs were not identified here and this warrants further investigation.

Overall, the 72 h supernatants of *A. fumigatus* isolates caused the greatest disruption to CFBE TJ integrity, and this effect was induced as early as 2 h post-treatment. *A. fumigatus* allergens and proteases have been reported to cause damage to epithelial barriers [35,39]. Amitani et al. investigated the effect of 7-day *A. fumigatus* supernatants on human nasal epithelial cells (HNCEs) from 14 different *A. fumigatus* clinical isolates [27]. They reported that *A. fumigatus* supernatants caused significant damage to HNCEs and slowed ciliary beat frequency [27]. However, the secretions of proteases are known to be minimal in the ATCC Af293 strain prior to day 4 of culture in MEM [40]. Considering that the 72 h supernatants in this study caused the most rapid and significant reduction in TJ integrity, we investigated whether gliotoxin, a secondary metabolite of *A. fumigatus* produced in highest quantities at 72 h growth [32,41], was capable of disrupting TJ integrity in CFBE monolayers. We found that commercially available gliotoxin at concentrations similar to those detected from BAL fluid of CF patients colonised with *A. fumigatus* [4] was capable of significantly reducing CFBE TER. Gliotoxin is known to have numerous immunoevasive effects including slowing ciliary beat frequency [27], inhibiting B and T lymphocyte proliferation [42] and hindering the ability of macrophages and neutrophils to phagocytose and kill microorganisms [43]. Gliotoxin has also been shown to significantly down-regulate the expression of the Vitamin D receptor in CFBE and HBE cell lines [4]. Khoufache et al. stated that gliotoxin isolated from *A. fumigatus* supernatants did not cause a decrease in the TEER of nasal epithelium [44]; however, we have shown that gliotoxin did significantly decrease the TEER of HBE and CFBE monolayers. To confirm the involvement of gliotoxin in disrupting CFBE monolayers, we employed a gliotoxin mutant strain of *A. fumigatus* which has been shown to be deficient in gliotoxin production by liquid chromatography–mass spectrometry (LC–MS) and reversed phase-high performance liquid chromatography (RP-HPLC) analysis of supernatants [31]. The 72 h supernatants of Δ*gliG* did not impair CFBE TJs, strongly suggesting that gliotoxin contributes to the breach of CFBE TJs caused by the 72 h supernatant of *A. fumigatus*. This has not been reported before. In fact, apart from one recent study showing that gliotoxin modulates actin cytoskeleton rearrangement to facilitate *A. fumigatus* internalisation into lung epithelial cells [45], there is a dearth of studies investigating the impact of gliotoxin on airway epithelia. Interestingly, we found that CFBE monolayers were more susceptible to gliotoxin than HBE monolayers. CF respiratory epithelia have been shown to respond differently than normal respiratory epithelium to microbial exposure [35] and it has been reported that the normal organisation and function of TJs is disturbed by the localisation of delF508-CFTR to the cytoplasm and retention within the endoplasmic reticulum [46]. Furthermore, CFTR has been shown to colocalise with ZO-1 at the TJ to regulate TJ assembly [47].

ZO-1 is a peripheral membrane protein found on the cytoplasmic surfaces of epithelial cell membranes and has been suggested to have both structural and signaling roles in the TJ [48]. JAM-A is a member of the immunoglobulin super family expressed in the TJs of epithelial cells and has been reported to play a role in leukocyte migration, cell–cell adhesion, homophilic interactions [25] and TJ barrier regulation [49]. Both TJ proteins are integral to the maintenance of epithelial barriers and their delocalisation is associated with TJ disruption. The western blot and confocal evidence presented here revealed that CFBE cells exposed to 72 h supernatants of Af293 had markedly delocalised ZO-1 and JAM-A. In contrast, CFBE cells exposed to the 72 h supernatants of the ΔgliG mutant exhibited no changes in ZO-1 and JAM-A levels and their localisation was similar to that of the control untreated monolayers. Given the previously reported abilities of gliotoxin to interfere with protein activity [50], and translocation [29,30], it is not surprising that gliotoxin can interfere with TJ proteins and epithelial barrier integrity. It is likely that the disruption of airway epithelial barriers by *A. fumigatus* is multifactorial with host factors playing a significant role in the ability of *A. fumigatus* to exert these effects in vivo. The breakdown of TJ integrity in the CF airway has obvious implications. Wan et al. demonstrated that a breakdown in TJ integrity in response to the house dust mite allowed migration of Der p allergens outside of the respiratory epithelium which could initiate an inflammatory response [51]. It has been hypothesised that disruption of airway epithelial TJ integrity could permit allergens access to the underlying capillaries [52,53].

The CF airway is host to a diverse microbiome [18] and interactions within the airway microbiome influence virulence, antimicrobial resistance and disease progression [54,55]. *A. fumigatus* and *P. aeruginosa* co-infections contribute to increased pathogenicity in a *Galleria mellonella* model and these co-infecting microbes contribute to an altered inflammatory response in CFBE cells [56]. Gliotoxin has been shown to cause apoptosis in human lung epithelial cells [57] and here we have shown that it can impair the integrity of the epithelial barrier. Therefore, damage of the epithelial barrier by *A. fumigatus* secretions could allow the translocation of *A. fumigatus* conidia (and other species in the microbiome), penetration of hyphae and the potential release of allergens into the blood stream which may exacerbate or initiate an allergic reaction such as ABPA. The evidence reported here, alongside the building body of evidence in the literature, reaffirms that *A. fumigatus* is not an innocent bystander in the chronic airway and that colonisation of the airways in the absence of ABPA has the potential to cause silent damage to the epithelium.

No studies have investigated the effects of *A. fumigatus* supernatants or gliotoxin on the integrity of bronchial epithelium with the delF508 mutation associated with CF. We report here that both reference and clinical strains of *A. fumigatus* can disrupt the TJs of epithelial monolayers. We show for the first time that *A. fumigatus* supernatants disrupt CFBE TJ integrity and we have provided evidence that gliotoxin in later supernatants is contributing to this effect. *A. fumigatus* can persistently colonise the CF airways (Appendix A) and gliotoxin has been detected in the airways of CF patients asymptomatically colonised with *A. fumigatus* [4]. Given the findings presented here, the exposure of CF airway epithelium to *A. fumigatus* secretions and gliotoxin can compromise the integrity of the epithelial barrier and potentially enhance the invasion of microorganisms in the lungs of people with CF.

## Figures and Tables

**Figure 1 jof-09-00490-f001:**
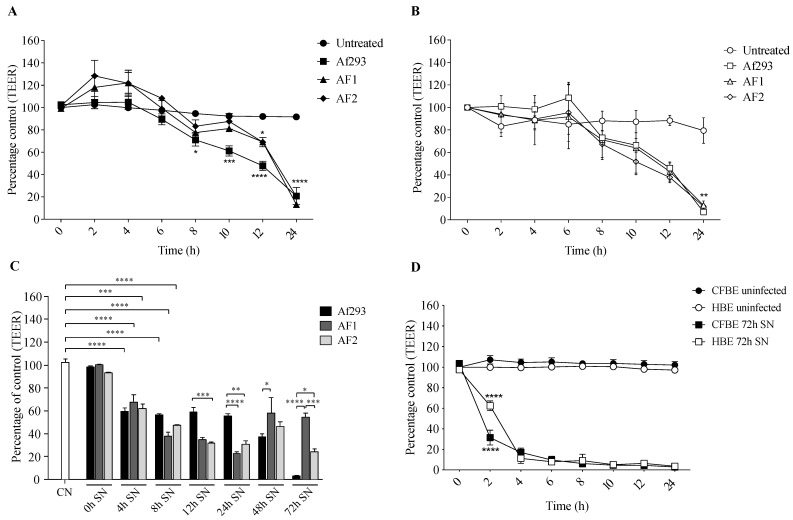
Integrity of CFBE and HBE monolayers following exposure to conidia or supernatants of *A. fumigatus*. (**A**) TEER values of CFBE monolayers untreated, infected with Af293 conidia, AF1 conidia and AF2 conidia at an MOI of 2:1. (**B**) TEER values of HBE monolayers untreated, infected with Af293 conidia, AF1 conidia and AF2 conidia at an MOI of 2:1. (**C**) Histogram presenting the percentage change from control TEERs of CFBE monolayers untreated (CN) and after 24 h exposure to 0, 4, 8, 12, 24, 48 and 72 h supernatants of Af293, AF1 and AF2. (**D**) TEER values of CFBE and HBE monolayers untreated and exposed to 72 h supernatants (SN) of Af293 over 24 h. Error bars represent standard error of at least three independent replicates. * *p* < 0.05, ** *p* < 0.01, *** *p* < 0.001, **** *p* < 0.0001.

**Figure 2 jof-09-00490-f002:**
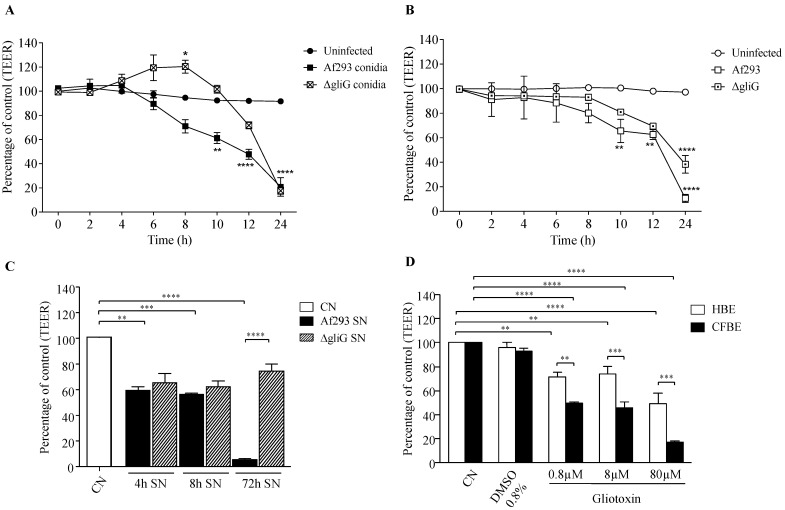
Effect of gliotoxin on the integrity of CFBE and HBE monolayers. (**A**) TEER of CFBE monolayers alone, exposed to Af293 conidia and Δ*gliG* conidia. (**B**) TEER of HBE monolayers alone, exposed Af293 conidia and Δ*gliG* conidia. (**C**) TEERs of untreated CFBE cells (CN, white bars) and CFBE cells following 24 h exposure to 4, 8 and 72 h supernatants of Af293 and Δ*gliG*. (**D**) TEER of CFBE and HBE following 24 h exposure to 0.08% DMSO, 0.8 µM, 8 µM and 80 µM gliotoxin. Error bars represent standard error of at least three independent replicates. * *p* < 0.05, ** *p* < 0.01, *** *p* < 0.001, **** *p* < 0.0001.

**Figure 3 jof-09-00490-f003:**
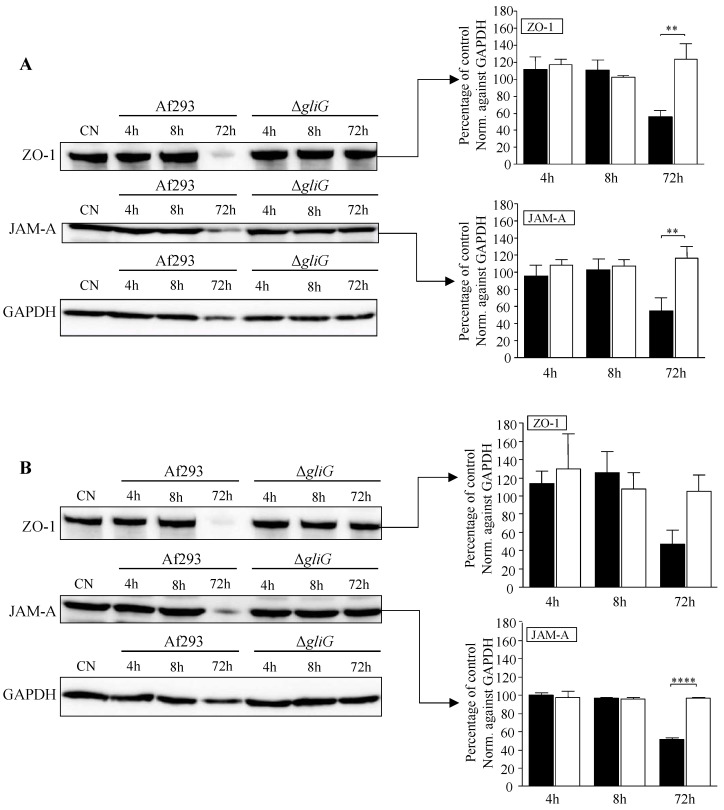
Western blot analysis of relative concentrations of ZO-1, JAM-A and GAPDH in CFBE and HBE monolayers following exposure to supernatants of *A. fumigatus* and Δ*gliG*. Western blots of ZO-1, JAM-A and GAPDH proteins from CFBE (**A**) and HBE (**B**) monolayers following 24 h exposure to culture medium alone (CN) or 4 h, 8 h or 72 h supernatants of Af293 or Δ*gliG*. Bar charts present densitometric analysis of ZO-1 (225 kDa) and JAM-A (36 kDa) bands from western blots to determine relative protein levels of monolayers exposed to Af293 supernatants (black bars) and Δ*gliG* supernatants (white bars). Band intensities from three independent experiments were normalised against GAPDH values and are expressed as the mean percentage of the control. Error bars represent standard error of at least three independent replicates. ** *p* < 0.01, **** *p* < 0.0001.

**Figure 4 jof-09-00490-f004:**
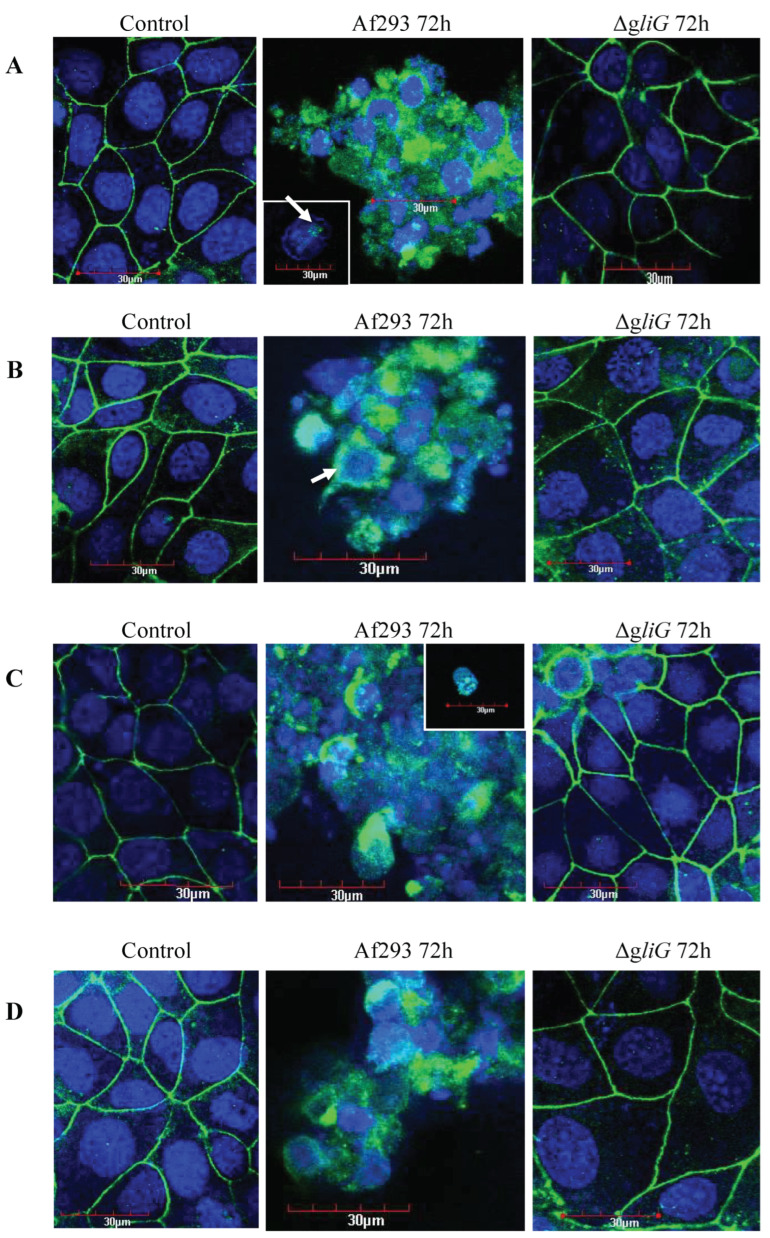
JAM-A and ZO-1 distribution in CFBE and HBE monolayers exposed to 72h supernatants of Af293 and Δ*gliG*. Confocal images at ×600 of (**A**) ZO-1 distribution and (**B**) JAM-A distribution in CFBE monolayers and (**C**) ZO-1 distribution and (**D**) JAM-A distribution in HBE monolayers following 24 h exposure to MEM media alone (control), 72 h supernatant of Af293 and 72 h supernatant of Δ*gliG*. ZO-1 and JAM-A were immunostained with FITC-conjugated anti-JAM-A and anti-ZO-1 antibodies (green). Monolayers were counterstained with DAPI to visualise the nucleus of the cells (blue). White arrows depicts abnormal distribution of ZO-1 and JAM-A within the cell cytoplasm. Scale bar indicates 30 µm.

## Data Availability

Data are contained within the article or Appendix A.

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
