# Peer review of "Aspergillus fumigatus Supernatants Disrupt Bronchial Epithelial Monolayers: Potential Role for Enhanced Invasion in Cystic Fibrosis"

_jof, 2023, doi:10.3390/jof9040490_

Round 1

Reviewer 1 Report

Manuscript would describe the role of A.fumigatus in CF. The subject falls within the scope of the Journal and the scientific content could be interesting and the paper is acceptable for publication. However, there are a number of points, relating to methodology, which need to be addressed.

Some Specific comments:

Introduction

Line 37 Please add here other microorganisms involved in CF and their role in CF. It may help readers understand why authors will cite Psudomonas and Burkholderia.

Line 43 clearance, this....

Line 57 Pseudomonas aeruginosa. Not abbreviate because it is the first time it is cited

Line 79 Please specify which effects.

In addition add why aspergillus infection is more important than other bacteria infection in CF

Line 92 How was the identification of Aspergillus fumigatus in clinical isolation done? The classification actually indicates the Aspergillus fumigatus complex, which includes several species that are similar to each other. Is the strain used by the authors a real A. fumigatus or one of the species of the A. fumigatus complex?

Although A. fumigatus is recognised as the major human pathogen within the complex, phylogenetic studies have demonstrated that some human and animal infections may be caused by A. lentulus, A. fumigatiaffinis, A. fumisynnematus, A. felis, etc. belonging to tha A.fumigatus complex.

Please specify

Line 237 A.fumigatus in italics

Lines 249-251 Conidia? It is not clear to me how conidia alone can be disruptive! Please explain better.

 References are not well presented; they should be described as instructions for authors guidelines.

Reviewer 2 Report

The manuscript presents a well-performed study giving much scientific value. The most interesting conclusions are the facts that A. fumigatus directly disturbs TJ of human BE cell lines (shown within 2 differenet TJ proteins - ZO-1 and JAM-A). Additionally, the Authors have shown that known Af toxin - gliotoxin is most likely contributing to this effect. 

Major question

In Fig 4 the Authors show the lack of effect of ΔgliG on TJ integrity. Also in discussion They stated that : "We show for the first time that A. fumigatus supernatants disrupt CFBE TJ integrity and we have provided evidence that gliotoxin in later supernatants is contributing to this effect.". Despite the fact that the effect of ΔgliG is rather obvious there is little possibility that some different effect might have occurred. In some cases deletion of a gene (and lack of certain gene product) might have a pleiotropic effect on the cell physiology and therefore lead to up- or downexpression of different genes, sometimes of a very different proccesses. Sometimes, the transformation method itself might cause genetic changes in the cells (example Bouchonville et al. Eukaryotic Cell 2009). The possible effect of gliotoxin on TJs is a very strong and important conclusion and it is worth a more in depth discussion on different possibilities which might have occurred in ΔgliG cells.

Minor questions

The Authors should check text editing. Examples:

In some parts of the ms A. fumigatus or ΔgliG are lacking italics.

Captions in different panels (such as in panel A, Fig1) should be unified to the same font size. Additionally, in panel B, Fig2 "percentage" should be capitalizied. 

The reference strain should be presented as "Af293" instead of "AF293".

Round 2

Reviewer 1 Report

The paper is available for publication in this form

Author Response

thank you

Reviewer 2 Report

I accept the current version.

Author Response

thank you